# The Importance of Technical Support in the Return of Traditional Crops in the Alps: The Case of Rye in Camonica Valley

**Valeria Leoni** [1,2] , **Davide Pedrali** [1] , **Marco Zuccolo** [1] , **Alessia Rodari** [1] , **Luca Giupponi** [1,2,*] and **Annamaria Giorgi** [1,2]

1   Centre of Applied Studies for the Sustainable Management and Protection of Mountain Areas-CRC Ge.S.Di.Mont., University of Milan, Via Morino 8, 25048 Edolo, Italy; valeria.leoni@unimi.it (V.L.); davide.pedrali@unimi.it (D.P.); marco.zuccolo@unimi.it (M.Z.); alessia.rodari@unimi.it (A.R.); anna.giorgi@unimi.it (A.G.)
2   Department of Agricultural and Environmental Sciences-Production, Landscape and Agroenergy-DiSAA, University of Milan, Via Celoria 2, 20133 Milan, Italy
*   Correspondence: luca.giupponi@unimi.it

**Abstract:** Multifunctional agriculture could be strategic for the recovery of some mountain areas of the Alps, and traditional crops like cereals generated study cases that triggered processes of development, such as rye in Camonica Valley (Northern Italy). However, farmers are often newcomers, and the specificities of low input agriculture make the training in agriculture fundamental. The impact of public workshops/seminars (organized by the Ge.S.Di.Mont. Research Centre of the University of Milan in Camonica Valley) on cereal cultivation between 2016 and 2021 was investigated. Moreover, rye produced in Camonica Valley was analyzed. The results show an increase in participation and a wider use of the streaming service. The percentage of participants not from an agricultural background had always remained about 50%, but decreased to 15.17% ± 5.07 in 2021, in contrast to the increase of professionals in agriculture and forestry. This is probably due to the accreditation of training activities for agronomists and foresters, and to the start of specific training projects regarding cereals. Samples of rye produced in Camonica Valley following the period of training activities were phytochemically/nutritionally characterized and compared to commercial rye. Locally produced rye proved to be comparable to the commercial one; however it showed a remarkable unevenness in secondary metabolite content and productivity, due to environmental differences and diverse agro-techniques.

**Keywords:** mountain areas; sustainable development; training in agriculture; cereals; rye; plant agro-biodiversity; Italian Alps

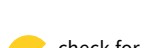



## 1. Introduction

European mountains are subjected to a parallel process of abandonment by population [1] and the arrival of newcomers. Therefore, the participatory process becomes more and more important in the conservation of biodiversity and agrobiodiversity, in particular in fragile territories where the intensive agricultural model is not applicable [2]. Agriculture is very often seen as a fundamental business for the recovery of abandoned land and the start of a process of recovery of marginal areas. A new phase has begun, over the course of a couple of generations, where a new awareness of the relationship with the earth has matured, and someone, strong in technology and knowledge that makes the difference, returns to look at the marginal areas and access to land as a fundamental element for promoting multifunctional agricultural activity [3].

Cereals are a typical agricultural product of the Alps since the Neolithic [4], and in recent times they were related to study cases of recovery of crops (e.g., rye) that triggered

virtuous processes of development framed in a general rethinking of mountains, in terms of living and doing business in accordance with local resources. Marginal areas are taking new meanings in the eyes of their inhabitants or of the people who decide to settle there from the city [5]. Very often, these new-born value chains follow the organic agricultural model. Organically produced foods are widely believed to promote a better environment and provide higher nutritive values. Several research studies conducted in many European countries have partly confirmed this belief, but also confirmed the lower productivity of organic farming, and also, with contradictory opinions, higher exposure to environmental contamination, and bacterial and fungi contamination (such as *Salmonella*, *Campylobacter*, mycotoxins, etc.) [6].

In Camonica Valley, different projects regarding the cultivation of cereals took place in the last few decades: in 2008–2010, the Province of Brescia started a project about minor cereals (rye, barley, and buckwheat) together with University of Milan (UNIMONT-Ge.S.Di.Mont. detached department) and the Study Centre on Mountains, setting up different experimental fields along the valley. In 2019, a Camonica Valley local bakery (Salvetti Forneria Pasticceria), started rye cultivation in the town of Malonno, a well-known area of rye cultivation until Second World War, and from 2018 to 2021, the Biodistrict of Camonica valley carried out the project: "Coltivare paesaggi resilienti" (cultivating resilient landascapes), financed by Cariplo foundation. This project started with the aim of counteracting the advancement of the forest and the abandonment of terraced and non-terraced arable land at altitudes between 500 m and 1500 m above sea level in Camonica Valley, and it involved a network of farms, local bodies, and institutions (schools and museums). Social cohesion and participation were thus considered a lever for improving the quality of life and an effective tool for safeguarding and maintaining the landscape. A symbolic cultivation, rye, was chosen as it could be directly employed to produce bread with the connected cultural aspects (traditional of the mountain territories, direct transformation, and production of a symbolic food for the development of proximity consumption). Among the objectives of the project, one important aim was the raising of the quality of farmers' skills through specialized training; thus, the participating members organized some workshops and seminars. In 2018, the Adamello Park (responsible for the technical monitoring activities related to the "Coltivare paesaggi resilienti" project) found a cereal disease in the fields, ergot caused by *Claviceps* spp., recognizing it as a possible food risk. The academic world was involved through the Ge.S.Di.Mont. Research Centre in the organization of a specific workshop on ergot (https://www.unimontagna.it/unimont-media/segale-cornuta-nei-cereali-alpini-un-tuffo-nella-storia-un-rischio-alimentare-attuale/, accessed on 23 November 2021) that was little known by new growers, as referred by the technicians of Adamello Park. This is probably due to the specific discontinuous nature of this disease [7], which highlights the training of farmers as a fundamental aspect in the fight against it and in maintaining food security. To increase the involvement of the citizenship, the folkloristic aspect was also considered; it should be remembered that alkaloids are also hallucinogenic drugs [8], therefore traditions and rituals of Camonica Valley could be linked to the ingestion of these toxic substances present in flours and their derivatives contaminated by ergot. For example, the consumption of foods contaminated with ergot may be linked to the famous witches' sabbath and the dancing of the jumping people of San Vito in Incudine, and some other famous folkloristic traditions of the area. The "storytelling" of local products and its traditions was useful in raising the attention of the citizenship.

The fact that farmers are often newcomers, coupled with the specificities of organic and low input agriculture, could then make the training in agriculture fundamental; although, the multiple stakeholders of the new aspect of agricultural activities and the phenomenon of hobby farming must also be taken into consideration.

In this framework, the Center of Applied Studies for the Sustainable Management and Protection of Mountain Areas (Ge.S.Di.Mont. Research Centre) (Edolo, Camonica Valley, BS, Italy) started an intense activity of research based on applied research focusing on

mountain specific challenges, centering on environment, agriculture, and forestry since 2006. The research activities were very soon accompanied by compelling activities of project planning and dissemination, delivered remotely almost from the start, with various multidisciplinary seminar initiatives, streamed through a virtual classroom and available on demand. The virtual classroom has proven useful in involving local communities and stakeholders, and to organize virtual worktables through social networks that encourage the exchange of experiences and the development of innovative processes, as well as the exchange of good practices.

The aim of this research work was to:

- Analyse the participation (number and age of participants, profession and modality of participation) to the training activities of the Ge.S.Di.Mont. Research Centre concerning cereal cultivation through seminar and workshop attendance from 2016 to 2021;
- Analyse the phytochemical/nutritional properties of rye produced in Camonica Valley in 2020 (comparing with commercial ryes) to understand if the production is acceptable in terms of health and safety.

## 2. Materials and Methods

### 2.1. Collection and Analysis of Data of Seminar Participation

The Ge.S.Di.Mont. Research Centre has been organizing and managing free seminars since 2012, in addition to the three-year bachelor's degree course in "Conservation and Sustainable Development of Mountain areas" (UNIMONT, www.unimontagna.it, accessed on 24 November 2021) of the Agricultural and Environmental Sciences Department of University of Milan, that was established in 1996. Seminars take place at UNIMONT, which is a detached center of University of Milan in Edolo (Brescia province) in Camonica Valley (an Alpine valley in the centre-east of Northern Italy; Latitude N 46°06′; Longitude E 10°20′) and are provided via streaming through a virtual classroom where the public can actively participate. Each activity is registered and made available on-demand (https://www.unimontagna.it/servizi/prodotti-e-servizi-multimediali/, accessed on 26 November 2021). In 2020–2021, the streaming virtual classroom was the only delivery possibile due to the COVID-19 emergency.

In the last six years, to be able to access the virtual classroom, participants were asked to provide some personal data as age (0–25, 26–40, 41–60, >60 years old), gender, profession (farmers, agricultural/forester technician, students, teacher/professor, other). Doing so, Ge.S.Di.Mont. was able to monitor the participants features and create a database of records.

Data were extrapolated from this database, considering only the seminars concerning cereals (growing, first transformation, local landraces etc.; the titles of each seminar are indicated in Supplementary Materials) realized between 2016 and 2021. Data were firstly divided per year, and different categories of data were considered:

1. Total participants (considering the percentage of participants in presence and in the virtual classroom);
2. Average participants for seminar (considering both in presence, in streaming and on demand);
3. Participants divided for age fascia (<40, which in Italy is considered "young" for the agricultural sector; >40);
4. Participants divided for profession (farmer, agricultural/forester technician, student, teacher/professor, other).

Finally, these categories of data were elaborated through Microsoft Excel spreadsheet considering media and percentages, on which linear graphics were built.

### 2.2. Plant Materials

Whole rye flour samples (identification code: 1–8; Table 1) from the productive season 2020 (Table 2) were collected directly from farmers. All the flour was obtained from rye



growing at different altitudes and in different sites of Camonica Valley (Figure 1). Detailed description can be found in Table 1. In addition, some ergot sclerotia (*Claviceps purpurea* (Fr.) Tul.) were collected from the sampling fields to be used in the ergot toxins analysis. Four samples of commercial whole rye flour (identification code: C1–C4; Table 1) were included as comparison.

**Table 1.** Varieties, origin, and cultivation altitude of rye samples. Key: PCV, produced in Camonica Valley; C, commercial.

| Sample ID Code | Cultivar | Origin | Municipality | Altitude (m.s.l.) |
|---|---|---|---|---|
| 1 | Dukato bio | PCV | Saviore dell'Adamello | 1200 |
| 2 | Conduct | PCV | Borno | 1000 |
| 3 | Population rye | PCV | Malonno | 560 |
| 4 | Dukato bio | PCV | Darfo Boario Terme (Gorzone) | 300 |
| 5 | Dukato bio | PCV | Corteno Golgi | 1000 |
| 6 | Landrace 1 | PCV | Corteno Golgi (Doverio) | 1100 |
| 7 | Landrace 2 | PCV | Edolo | 1100 |
| 8 | Diamond | PCV | Pian Camuno | 250 |
| C1 | - | C (German) | | - |
| C2 | - | C (Italian) | | - |
| C3 | - | C (England) | | - |
| C4 | - | C (Austria) | | - |

**Table 2.** Cultivated area and production data of PCV (produced in Camonica Valley) rye during the year 2020.

| Farm/Sample | Cultivated Area (ha) | Production (kg) | Production (q/ha) |
|---|---|---|---|
| 1 | 0.045 | 20 | 4.44 |
| 2 | 0.316 | 500 | 15.82 |
| 3 | 0.600 | 400 | 6.67 |
| 4 | 0.073 | 120 | 16.44 |
| 5 | 0.270 | 670 | 24.81 |
| 6 | 0.025 | 45 | 18.00 |
| 7 | 0.030 | 90 | 30.00 |
| 8 | 2.000 | 4800 | 24.00 |
| Total | 3.360 | 6645.00 | - |
| Average | 0.420 | 830.63 | 17.52 |

Around 1000 g of each flour sample were collected in a plastic zip pouch and stored in the laboratory at room temperature in the dark prior the analysis.

*2.3. Phytochemical Analysis*

2.3.1. Chemicals and Reagents

All the chemicals and reagents were of commercial grade purity, and were purchased from Merck (Milan, Italy).

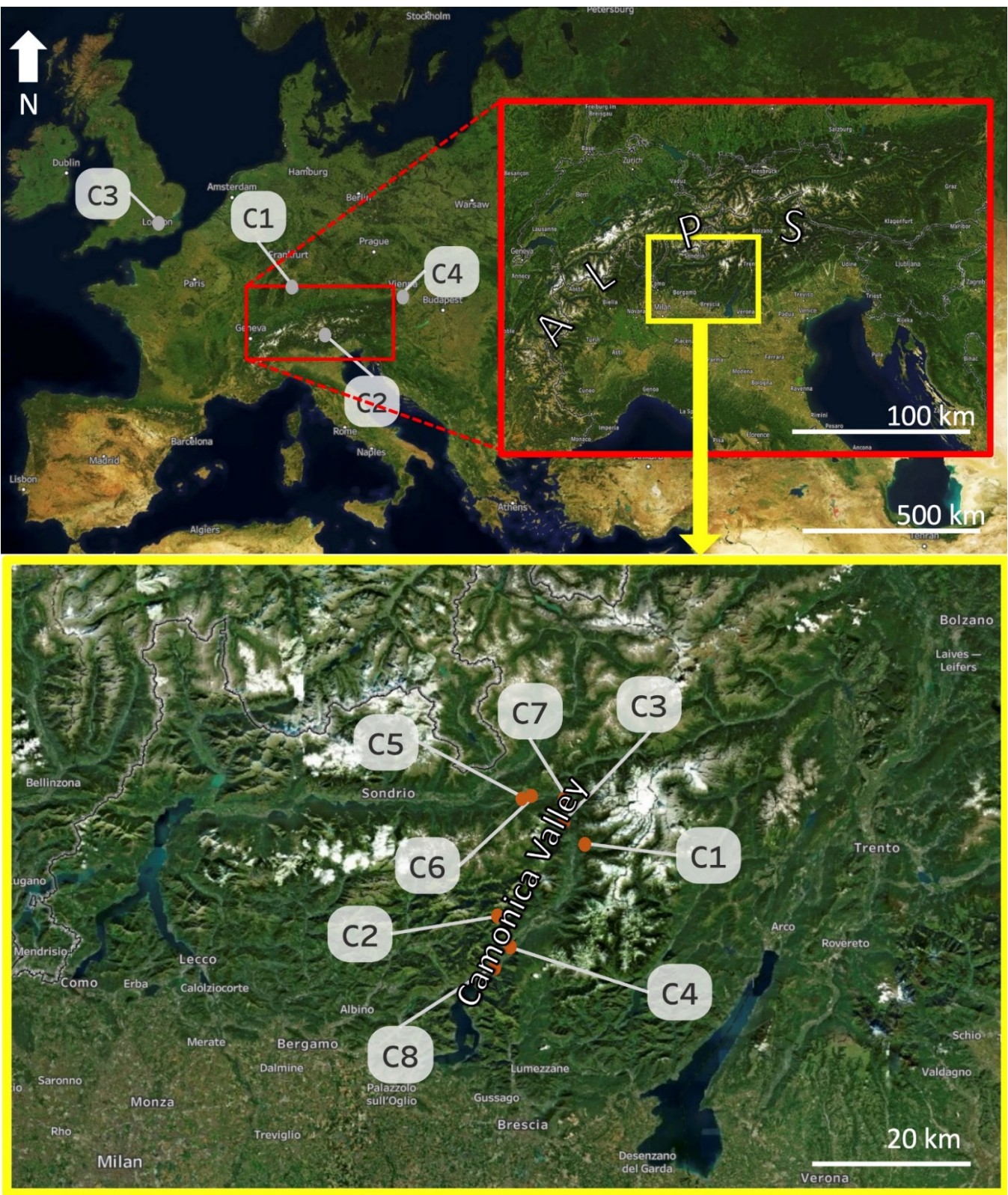

**Figure 1.** Geographical provenience of the rye samples. Red square shows the Alps, while the yellow one shows Camonica Valley. The identification code of the samples is the same as that used in Table 1.

### 2.3.2. Proximate Analysis

The moisture content of the whole rye flour was determined according to the AOAC Method No.945.15 [9]. The total nitrogen content was determined by the Kjeldahl method according to the AOAC Method No.945.18 [10]. The ash content was determined according to the AACC Method No.08-01.01 [11]. Total carotenoid content was determined according to the AACC Method No.14-60.01 [12]. The results were expressed in micrograms of lutein equivalents per gram of dry weight (µg lutein EQ/g dw). All the assays were carried out in triplicates.

### 2.3.3. Extraction and Analysis of Phenolic Compounds

Extraction of the phenolic compounds has been performed, adapting the procedure reported by Kulichova et al. [13]. Briefly, whole rye flour (10 g) was extracted with 100 mL of 80% methanol (*v*/*v*) containing 0.5% of formic acid (*v*/*v*) by stirring overnight in the dark. After that, the supernatant was collected by centrifugation (Hermle z300, HERMLE Labortechnik GmbH, Wehingen, Germany), and the residue was resuspended in 50 mL methanol and stirred for further 50 min. The combined supernatants were evaporated to dryness by a rotary evaporator at 50 °C (LABOROTA 4000eco, Heidolph InstrumentsGmbH & Co., Schwabach, Germany). The residue was dissolved in methanol and the final volume was adjusted to 2 mL. The resulting solutions were stored in the dark at −20 °C for up to 24 h. The samples that were needed were filtered through a 0.22 µm syringe filter.

The total polyphenol content (TPC) was determined using the Folin–Ciocâlteu method, adapting the procedure reported by Saura-Calixto and Bravo [14]. Briefly, 0.2 mL of extract were mixed with 0.2 mL of Folin's reagent and diluted with 3 mL of water. Then, 0.75 mL of 7% aqueous sodium carbonate (*w*/*v*) was added, and the resulting mixture was incubated for 8 min at room temperature. Finally, 0.85 mL of water was added, and the samples were stirred in a vortex mixer and incubated in the dark for 2 h at room temperature. The absorbance was measured at 765 nm against a reagent blank (Varian Cary 50 scan, Agilent, 5301 Stevens Creek Blvd, Santa Clara, CA, USA). All the assays were performed in triplicate. The results were expressed in milligrams of gallic acid equivalents per gram of dry weight (mg GAE/g dw).

The total flavonoid content (TFC) was determined using the aluminum chloride colorimetric assay, adapting the procedure reported by Alcazàr-Valle et al. [15]. Briefly, 0.05 mL of extract was diluted with 0.7 mL of water and mixed with 0.25 mL of a solution prepared by dissolving 133 mg of aluminum chloride and 400 mg of sodium acetate in 100 mL of methanol/water/acid acetic 140:50:10 (*v*/*v*/*v*). The absorbance was measured at 415 nm against a blank reagent. All the assays were performed in triplicate. The results were expressed in milligrams of quercetin equivalents per kilogram of dry weight (mg QE/kg dw).

The total anthocyanin content was determined accordingly to the procedure reported by Abdel-Aal and Hucl [16]. Briefly, whole rye flour (3 g) was extracted with 24 mL of absolute ethanol containing 0.15% of 0.1 M hydrochloric acid (*v*/*v*) by stirring for 30 min. After that, the supernatant was collected by centrifugation and the volume adjusted to 50 mL with the same solvent used for the extraction. The absorbance was measured at 535 nm against a reagent blank. All the assays were performed in triplicate. The results were expressed in milligrams of cyanidin 3-glucoside equivalents per kg of dry weight (mg Cy3G/kg dw).

### 2.3.4. Antioxidant Activity

The antioxidant activity was determined using the DPPH (2,2-diphenyl-1-picryl-hydrazyl-hydrate) free radical scavenging method, adapting the procedure reported by Brand-Williams et al. [17]. Briefly, 0.3 mL of the whole rye extract were mixed with 2.7 mL of a $6 \times 10^{-5}$ M methanol solution of DPPH free radical. The resulting solution was incubated for 30 min in the dark at room temperature. A DPPH blank was prepared in the same way using 0.3 mL of methanol. Finally, the absorbance was measured at 517 nm



against methanol as a blank. The antioxidant activity was calculated as the percentage of RSA (radical scavenging activity) using the formula RSA = [(AB − AA)/AB] × 100, where AB is the absorbance of the DPPH blank, and AA is the absorbance of the sample solution.

### 2.4. Ergot Toxins Analysis

The analysis of the ergot toxins was performed, adapting the procedure reported by Sulyok et al. 2006 [18]. Briefly, the whole rye flour (1 g) was extracted with 4 mL of acetonitrile/water/acetic acid 79:20:1 ($v/v/v$) by stirring for 90 min at room temperature. After that, the supernatant was collected by centrifugation and 0.35 mL was mixed with 0.35 mL of acetonitrile/water/acetic acid 20:79:1 ($v/v/v$). The samples that were needed were filtered through a 0.22 μm syringe filter. The same procedure was used to extract a sample of ergot sclerotia (1.4 g). The extract was diluted 10-fold with the same solvent system before the injection.

Ergocristine was used as reference standard. A stock solution at 1 mg/mL was prepared dissolving 2 mg of ergocristine in 1 mL of acetonitrile/water/acetic acid 79:20:1 ($v/v/v$). The resulting solution was diluted with 1 mL of acetonitrile/water/acetic acid 20:79:1 ($v/v/v$) and stored at 5 °C. The working standard mixture solutions were made by diluting with the same solvent system the stock standard solution to obtain two solutions at the concentrations of 100 and 10 μg/mL.

The High-Performance Liquid Chromatography (HPLC) system used was a LC Agilent series 1200 (Waldbronn, Germany) consisting of a degasser, a quaternary gradient pump, an auto-sampler and a MWD detector (Waldbronn, Germany). A Gemini® 5 μm C18 (150 × 4.6 mm) column (Phenomenex, Santa Clara, USA) at 25 °C was used for this analysis. Sample injections were made at 10 μL for all samples and standards. The run time was 20.05 min, with no post run time. A binary gradient comprising of water/methanol 90:10 ($v/v$) containing 5 mM ammonium acetate and 1% acetic acid ($v/v$) (A) and water/methanol 10:90 ($v/v$) containing 5 mM ammonium acetate and 1% acetic acid ($v/v$) (B) at a flow rate of 1.0 mL/min was used as the mobile phase. The gradient profile was as follows: 0 min, 100% A; 14 min, 100% B; and 20.05 min, 100% A. Absorbance wavelength was 310 nm, reported to be close to the max. absorption of ergopeptine alkaloids [19].

### 2.5. Statistical Analysis

Results were analyzed using one-way analysis of variance (ANOVA) (the assumptions of normality of group data and homogeneity of variances had been verified using the Shapiro–Wilk test and Levene's test, respectively), with Tukey test applied post-hoc, using SPSS Statistics 24.0 software. The data were expressed as mean ± standard deviation (SD) and differences were considered statistically significant when $p < 0.05$. In addition, the samples were ordered in multi-dimensional space using principal component analysis (PCA) performed by Statgraphics 5.1 (STCC Inc.; Rockville, MD, USA).

## 3. Results

### 3.1. Participation in Seminars

The graphs in Figure 2 show the participations in seminars from 2016 to 2021.

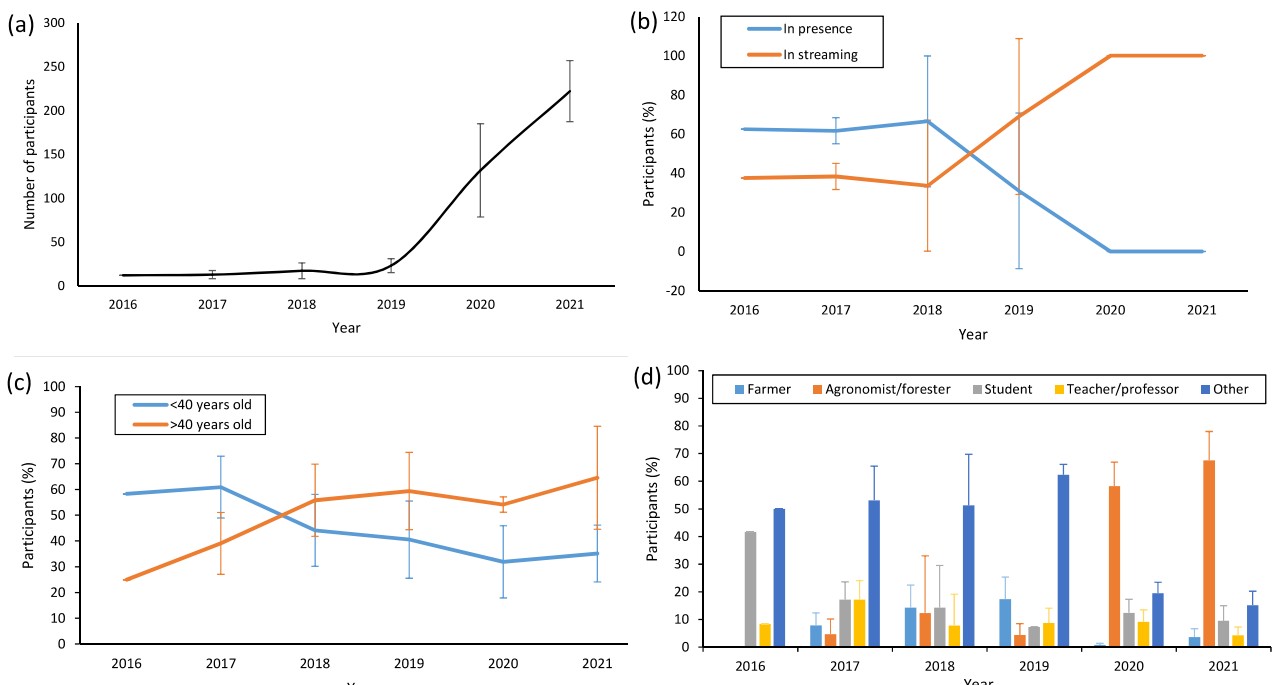

**Figure 2.** Participation in seminars: (**a**) total number of participants per year (from 2016 to 2021); (**b**) modality of participation (in presence or in streaming) per year; (**c**) age of participants; (**d**) profession of participants.

It is remarkably evident how the average number of participants for the event grew since 2019, reaching an average of 222 ± 34.87 participants per seminar. The percentage of participants in presence remains similar until 2018, when there is an inversion of the trend; from then on, there is an important increase in the use of the virtual room until reaching, in 2020, 100%.

In graph c (Figure 2), participants are divided based on the age (older than 40 and younger than 40). It is possible to evidence how the percentage of "young" participants (<40 years old) decreased from 60.94% ± 12 in 2017 to 31.90% ± 14 in 2020. In 2021, the percentage slightly re-increased (35.14% ± 11). Conversely, it is possible to see an increasing of the participants >40 years old, passing from 25% in 2016 to 64.56 ± 20% in 2021, except a slight decrease in 2020 (54.18 ± 3%). From Figure 2, it is also possible to see how in the first year under consideration (2016), the percentage of <40 years old (58.33%) is clearly higher than >40 years old (25%). The inversion of the trend happened in 2017.

From the analysis of the profession of participants, it is possible to see how students decreased from 41.67% in 2016 to 7.25% in 2019; a slight increase happened in 2020–2021, but never went higher than 10%. Farmers participating to seminars increased from zero to 17.39% ± 7.95 in 2019, decreasing again in the following years. Conversely, the participation of Foresters and Agronomists increased, passing from zero to 67.57% ± 10.39 in 2021. Professors and teachers remained quite consistent in the years considered, with a peak of 17.19% ± 6.88 in 2017. Lastly, the percentage of "other" (including all the people with a profession far from the agricultural or formative world) remained always higher compared to students, agronomists/foresters and teachers in the period considered (about 50% of participants), but decreased to 15.17% ± 5.07 in 2021, conversely to the increase of professionals in the agricultural field (as said above).

### 3.2. Quality of Cereal Supply Chain

The nutritional and phytochemical composition of the rye samples are presented in Table 3 and in Figure 3. Generally, no significant differences were observed between the produced in Camonica Valley (Produced in Camonica Valley—PCV) and commercial (Commercial—C) rye (Table 2), but inside the PCV there was higher variability.

**Table 3.** Student t-test analysis of nutritional and phytochemicals parameters of rye.

| Factor | Mean PCV Rye (*n* = 8) | Mean Commercial Rye (*n* = 4) | *t*-Value (df = 10) | *p*-Value |
|---|---|---|---|---|
| Moisture (%) | 10.29 ± 0.13 | 10.30 ± 0.19 | 0.0270 | 0.9790 |
| Protein (g/100 g) | 7.95 ± 0.21 | 8.82 ± 0.26 | 1.5782 | 0.1456 |
| Ash (%) | 0.77 ± 0.03 | 0.69 ± 0.01 | 0.3549 | 0.7300 |
| TCC (g lutein EQ/g) | 11.06 ± 0.07 | 15.85 ± 0.08 | 1.1156 | 0.2907 |
| TPC (mg GAE/g) | 24.37 ± 0.78 | 28.75 ± 0.96 | 0.4677 | 0.6500 |
| TFC (mg/kg) | 85.58 ± 0.26 | 82.75 ± 0.2 | 0.4694 | 0.6489 |
| Anthocyanin (mg Cy3G/kg) | 1.51 ± 0.19 | 1.41 ± 0.15 | 1.8350 | 0.0997 |
| DPPH (%) | 9.13 ± 0.1 | 6.65 ± 0.00 | 1.2245 | 0.2488 |

df: degrees of freedom. Protein, ash, TCC, TPC, TFC, anthocyanin and DPPH are referred to dry weight.

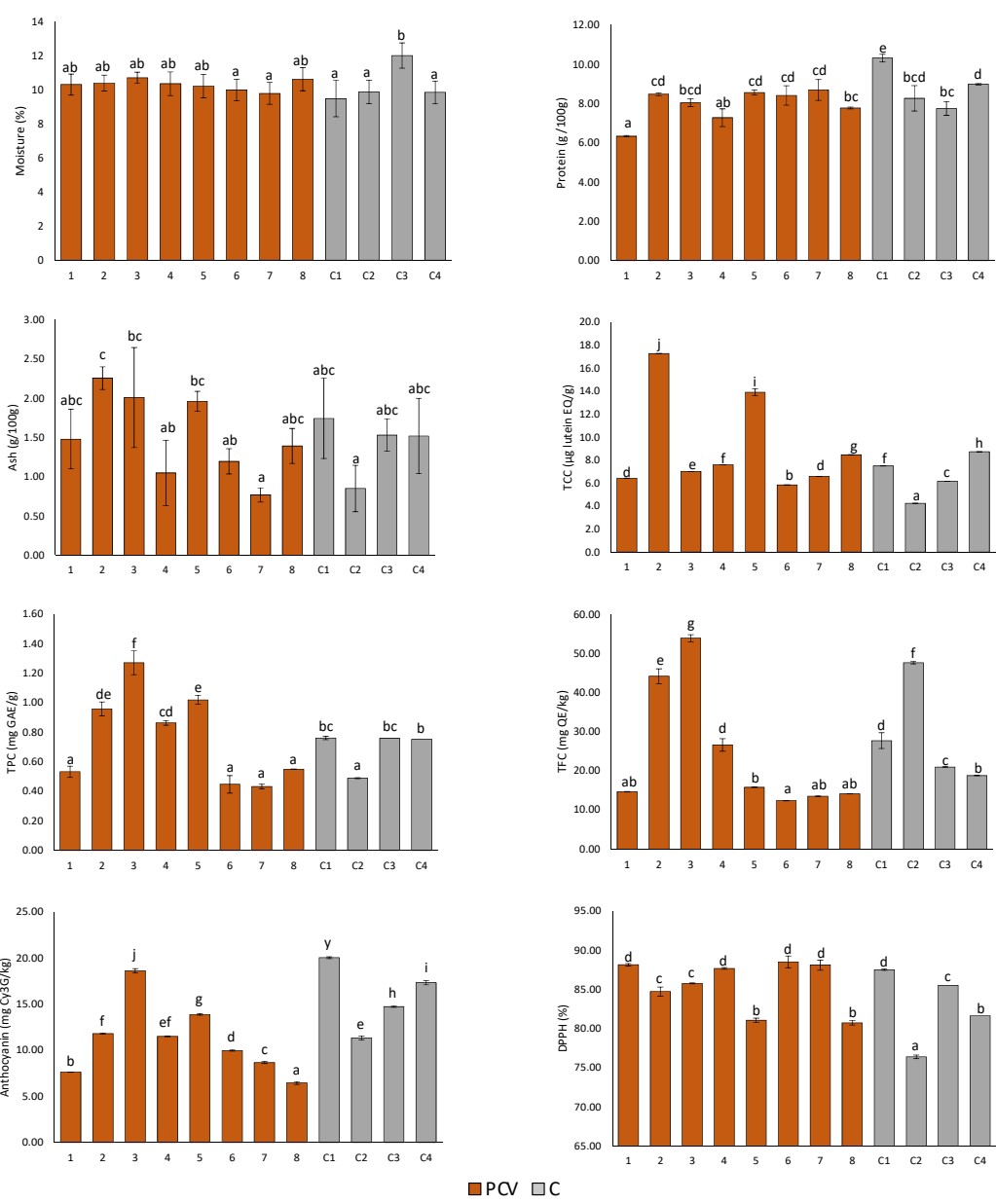

**Figure 3.** Nutritional and phytochemical features of rye flour (dry matter). The identification codes of the samples are the same used in Table 1 (PCV, produced in Camonica Valley; C, commercial). Data in the same column with different superscript letters are significantly different, *p* < 0.05.

Moisture, ash, and protein content were homogeneous among the samples, with few exceptions (C1 showed the highest protein level of 10.31% ± 0.2). A greater variability was observed for Total Carotenoid Content—TCC, Total Phenolic Content—TPC, and Total Flavonoids Content—TFC, with PCV rye samples showing a larger unevenness than commercial ones. Particularly, PCV 2 and PCV 5 had the highest TCC content, with 17.25 ± 0.02 µg lutein EQ/g (DW) and 13.90 ± 0.30 µg lutein EQ/g (DW), respectively, whereas the PCV 3 exhibited the greatest levels of TPC and TFC (1.27 ± 0.08 mg GAE/g and 53.84 ± 0.91 mg/kg (DW)). Generally, the commercial rye was higher in anthocyanin content than PCV samples, with the exception of PCV3 having 18.61 ± 0.24 mg Cy3G/kg (DW).

Regarding the antioxidant activity in the DPPH (2,2-diphenyl-1-picrylhydrazyl) assay, no general trends emerged, with all the samples showing a similar percentage of inhibition.

These results were confirmed by PCA biplot (Figure 4). The first two principal components (PC) explain 61.79% of total variance (PC1 = 43.23%; PC2 = 18.56%). The commercial samples were situated in the middle of the graph where protein and anthocyanin were positioned, while the PCV rye was more spread in the biplot region. The sample PCV 2, PCV 3 and PCV 5 were located to the right in the score plot, like TCC and TPC.

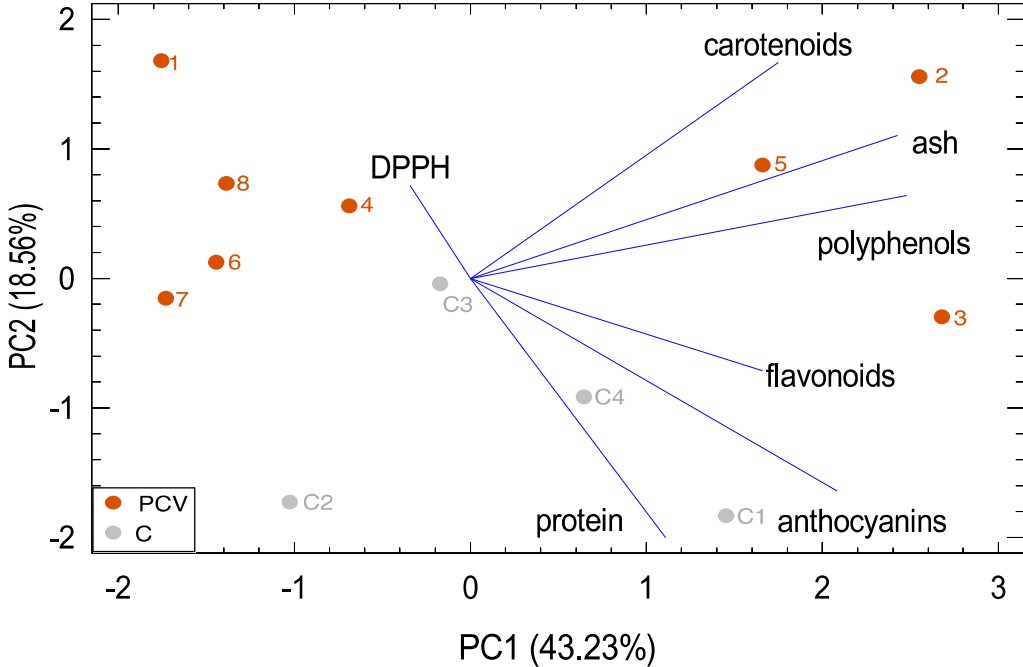

**Figure 4.** Principal component analysis (PCA) biplot of rye samples associated with nutritional/phytochemical variables.

Figure 5 displays the HPLC chromatograms of rye extracts. Both C and PCV samples show similar chromatographic profiles. All the runs show high similarity, with few differences related to the kind and the signal intensity. An ergot extract and ergocristine solution were used as a standard; no signals associated with ergot alkaloids have been observed in all the rye extracts. Interestingly, the ergocristine is not present in the ergot extract chromatogram.

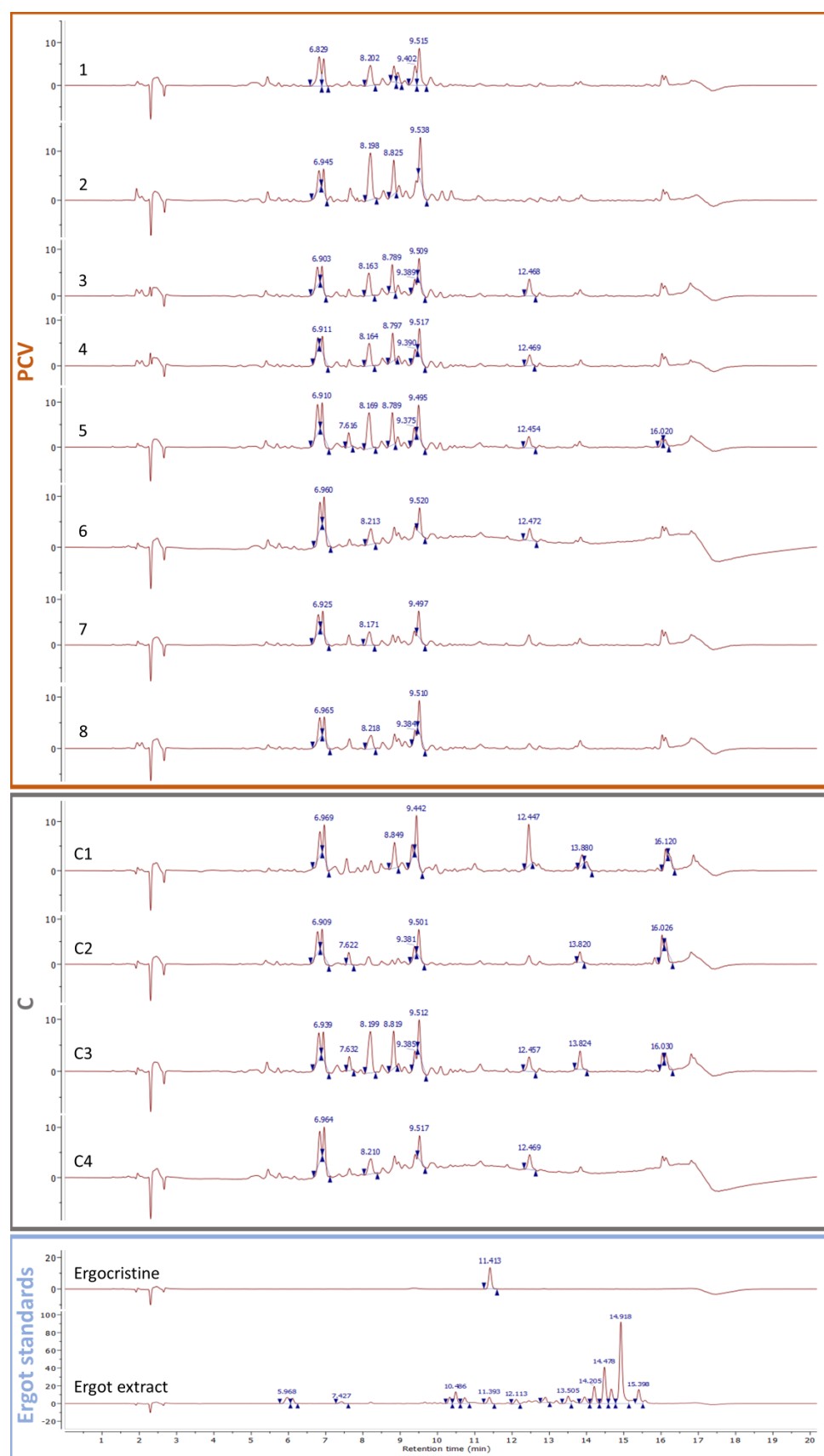

**Figure 5.** Comparison of HPLC chromatograms of rye samples (PCV, produced in Camonica Valley; C, commercial), ergot extract, and ergocristine standard.

## 4. Discussion

### 4.1. Participation in Seminars

Considering the participation in seminars (Figure 2), it is possible to see how the percentage of "non-professionals" (the class "other") is remarkable in the period considered; consistently almost 50% until 2020, when it decreases to approximately 20% of participants, while the "professionals in the field" (agronomists and foresters) increases. This last changeover could be due to the fact that, from 2020, the seminars became accredited training for agronomists and foresters. These categories are required by Italian law to follow Continuing Education Courses for a certain number of credits (indicated as CFP, corresponding to eight hours of formative activities) from 2013, to facilitate professionals' adaptation to technological, organizational, and contextual changes. Further, the growth of the participants in general testifies how the subject is receiving increased attention from the public, and confirms that interest in sustainable agriculture has grown in the last decades [20].

As mentioned, there is a new awareness towards agriculture by the non-professional public. In Alpine valleys, as in other parts of the world, the people returning to mountains are often newcomers, or they skip one or two generations. There are also multiple successors who worked away from the country, gaining sufficient wealth or income that they returned to farming on a non-commercial basis or semi-commercial basis [21]; in part-time farming, most farmers obtained stable off-farm jobs, then added farming later in their adult years [21]. We can add hobby farming to these processes, which is recognized as a primary category in recent farming typologies [22]. Hobby farming is typically described as small-scale and undertaken on a part-time basis by exurbanites interested in commodity production as a recreational activity [23]. In addition, these "new" farmers seek a social status not only for the ownership and occupation of agricultural land, but also through membership to desirable groups evaluating further cultural symbols, such as organic products and pedigree cattle [24], or maintaining a harmonic landscape, as it is happening in Camonica valley with the project "Resilient landscapes". These trends coincide with the recent agricultural policies that move towards the encouragement of environmentally friendly farming and the greening of agricultural policies [25]. The European Green Deal foresees a more sustainable agriculture maintaining biodiversity and soil health. The increase of organic production is a part of the deal, which is indicated in the Farm to Fork and Biodiversity Strategies as the objective of 25% of the EU's agricultural land under organic farming by 2030 [26]. This has been transposed at a national level with the PNRR (Recovery and Resilience National Plan), which requires a green revolution and ecological transition of Italian economy, and hence of agricultural activities, setting out actions for sustainable agriculture. In this framework, the reviewing process revenue model for academic staff is also undergoing, to provide continuity to the role of academy for rural and society development.

Contemporary agriculture and rural society can be defined as a multifunctional agriculture regime [27] operating at various scales (recreation, leisure, environmental conservation, re-establishment of lost or damaged habitats). In modern agricultural systems, there is a shift toward sustainable agriculture [28] and the replacement of physical input on farm with knowledge inputs [29]. The European Green Deal also forecasts the goal of "leaving no one behind" [26] and it is possible to see from the results how the connection (virtual and physical) of the marginal territories is important for their development.

European goals naturally overlap with United Nations' Sustainable Development Goals, and proper training for a sustainable agricultural development of marginal areas matches a number of the seventeen SDG, for example SDG 4 (Quality Education), and SDG 12 (Responsible Consumption and Production), and the participatory process with local communities coincides with SDG 17 (Partnership for the Goals) [30].

The trends described above, together with the necessity of continuing education for agronomists and foresters, could also explain the inversion of the trend in the age of participants, with over-forties reaching more than 60% in 2021. Young participants

are present in all the periods considered, but at a lower percentage in the more recent years. Previous seminars were live-streamed and recorded, but most were attended mainly by students of the Faculty of Agronomy, while, from 2018, there was an apparent turnout of the public, maybe also due to the start of two important projects concerning cereals cultivation, the already described "Resilient landscape", but also the CereAlp project ("Good practices for the cultivation and processing of alpine cereals and medicinal plants": https://www.unimontagna.it/progetti/buone-pratiche-per-la-coltivazione-e-la-trasformazione-di-cereali-alpini-e-piante-officinali-ceralp/, accessed on 24 November 2021); a project from the Lombardy region concerning activities for the dissemination of knowledge about cereal (and pseudocereal) landraces and officinal plants currently cultivated in Lombardy. The results testify the importance of regional planning and how intermediate and higher education in agriculture plays a decisive role in rural development and sustainable agricultural production [31,32]. Higher Education entities need to pass from hierarchical organizations to participatory ones, and shift from agricultural universities to universities for rural development [33].

A further increase in the number of participants happened in 2019, when the pandemic due to COVID-19 and social distancing prompted a rapid development and implementation of a complete virtual education. The virtual classroom was already a fundamental tool to reach a wide public from marginal locations in the Italian Alps, but the virtual mode proved indispensable, giving everyone the opportunity to inform themselves during the lockdown phase. As in other cases, COVID-19 also proved to be a way of reformulating education [34]. An increase of the virtual participation was in fact already happening in the two previous years, but reached 100% virtual participation during the pandemic, in 2020.

### 4.2. Quality of Cereals Supply Chain

From the nutritional point of view, and also for the presence of mycotoxins, commercial and local ryes were comparable. However local rye showed remarkable unevenness regarding secondary metabolites. Age of the plant, season, microbial attacks, grazing, radiation, competition, and nutritional status have been proven to have an impact on the secondary metabolite profile in higher plants, which is then influenced then by many and different factors [35,36]. Apart from the 2-ha field of sample eight (Table 2), the other samples were from small plots at different altitudes and cultivated with different agro-techniques, and this could have led to this unevenness.

The difference in the agro-technique and variety of the seeds is also testified by the unevenness in the harvest, ranging from 4 to 30 q/ha, while the yield is fixed between 18 q/ha and 35 q/ha in the literature [37]. Adamello Park technicians also refer to the very different costs of production reported by farmers [38], depending on the efficiency of agro-techniques, in particular concerning mechanization. Apart from the different varieties, then, the diversification of yield was linked to altitude, time of sowing, composition of the soil, and defense of the crop from grazing of wild and domestic animals. From the disparity of harvest, it is clear that some farms demonstrate good production capacity, comparable to the yield range stated in the literature. These situations seem to confirm the ability of mountain farms to respond to the challenge of the return to agriculture in marginal areas [39–42]. The less productive farms must refer to the more productive ones as valuable study cases to improve their productivity; therefore, it is very important to create a network among farmers and project initiatives inside local communities, training, and didactic activities, and to favor moments of confrontation among farmers. As referred to by Adamello Park technicians, at the end of the project, with the increased collaboration of farmers and more efficient work, it was possible to reach a visible (although still in low quantities) cereal/flour production within the market. The increase in yields can be attributed to an improvement in agro-techniques due to training activities, as well as the purchase of suitable mechanized means for cultivation in mountain areas within the project (seeder and combine harvester crawler) and the use of varieties less susceptible to bedding and with more uniform spikes. Adamello Park technicians also report a significant increase

in field productivity (from 12 q/ha in 2018 to 19 q/ha on average in 2020, with a very wide range of 4 to 25 q/ha [38]).

The Adamello Park monitored all farms involved in the project, through inspections and the compilation of field notebooks, and ensuring continuous technical support. Data sheets and periodical bulletins were created to sort out specific problems encountered, such as *Claviceps purpurea*, against which the Adamello Park acted in cooperation with the Ge.S.Di.Mont. Research Centre, confirming the importance of education and technical support in agriculture. The study case confirmed the lower productivity of organic farming; however, although some authors reported a higher exposure to contamination of organic produced food [6], concerning mycotoxin presence in the samples analysed, the local produced rye proved to be comparable with the commercial ones, with both C and PCV samples showing similar chromatographic profiles, confirming the efficacy of the technical support and formative activity.

## 5. Conclusions

This work considered the importance of education and technical support in the re-turn of traditional agricultural models in marginal areas, such as the Alps, for their sustainable development. It also analysed the public interest in teaching and training activities regarding cereals, and examined a particular study case, the production of rye, and also analysed its nutritional features. The production of cereals in the Alps is possible and generated a product with characteristics comparable with commercial rye, but the study revealed the necessity of formative activity and technical support, and universities can play a crucial role in rural development. The results testify the importance of regional planning and how intermediate and higher education in agriculture plays an important role in rural development and sustainable agricultural production. Furthermore, the results highlight the need to work in close relationships with land managers and in general with the stakeholders of the mountain marginal territories for their sustainable development.

**Supplementary Materials:** The following are available online at https://www.mdpi.com/article/10.3390/su132413818/s1, Table S1: List of seminars/workshops, organized by the Ge.S.Di.Mont. Research Centre of the University of Milan, in Camonica Valley, from 2016 to 2021.

**Author Contributions:** The first four authors contributed equally. Conceptualization, V.L., D.P., L.G. and A.G.; methodology, V.L., D.P., M.Z. and A.R.; validation, M.Z., L.G. and A.G.; formal analysis, V.L., D.P., M.Z., A.R. and L.G.; investigation, V.L., D.P. and A.R.; resources, D.P. and A.R.; data curation, V.L., D.P., M.Z., A.R. and L.G.; writing—original draft preparation, V.L., D.P, M.Z. and A.R.; writing—review and editing, V.L., D.P., M.Z., A.R. and L.G.; supervision, L.G. and A.G.; project administration, A.G.; funding acquisition, A.G. and L.G. All authors have read and agreed to the published version of the manuscript.

**Funding:** This research was supported by the "Montagne: Living Labs di innovazione per la transizione ecologica e digitale", "CereAlp- Buone pratiche per la coltivazione e la trasformazione di cereali alpini e piante officinali" and by "MIND FoodS HUB: Innovative concept for the eco-intensification of agricultural production and for the promotion of dietary patterns for human health and longevity through the creation in MIND of a digital Food System Hub" projects.

**Institutional Review Board Statement:** Not applicable.

**Informed Consent Statement:** Not applicable.

**Acknowledgments:** We wish to thank Guido Calvi, policy officer-agronomist at Comunità Montana di Vallecamonica, Unit "Servizio Parco Adamello e tutela ambientale", for sharing the data of the "Paesaggi resilienti" project and for his cooperation. Finally, we want to thank the rye's producers, "Biodistretto di Valle Camonica" and "Forneria Pasticceria Salvetti", for their collaboration.

**Conflicts of Interest:** The authors declare no conflict of interest.

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
