# Peer review of "The Importance of Technical Support in the Return of Traditional Crops in the Alps: The Case of Rye in Camonica Valley"

_sustainability, doi:10.3390/su132413818_

Round 1
Reviewer 1 Report
This is an interesting study comparing commercial versus small scale rye production in the Alps, as well as giving some information on the role of training seminars and who participates. It is well written and the figures and tables are of decent quality. The paper needs some minor English editing (the word 'the' does not belong in fronton a percentage in English), but overall it is a worthy paper. I also recommend the authors include a map showing the Alpine region, and a large-scale insert the study region.
Reviewer 2 Report
The attempt to show the impact of the university on local sustainability is appreciable. The authors have considered the genuine gaps of people leaving the agricultural profession and parallelly others venturing into this with the organic model. Both the objectives are served with proper work and justification throughout the paper. The use of tools like SPSS and techniques like PCA will speak for the quality of the data analysis.
The authors have a good grip over the research and have shown their maturity throughout the paper. However, the following are a few observations:
- Here and there compound sentences make it less smooth to navigate through the flow of thoughts.
- Mapping to the sustainable Developmental goals in the conclusion would be appreciated.
- The revenue model for the university team needs to be investigated as they also have started programs related to the intervention.
Excellent selection of the problem and great justification for choosing it.
